# Adeno-Associated Virus-like Particles’ Response to pH Changes as Revealed by nES-DMA

**DOI:** 10.3390/v15061361

**Published:** 2023-06-13

**Authors:** Samuele Zoratto, Thomas Heuser, Gernot Friedbacher, Robert Pletzenauer, Michael Graninger, Martina Marchetti-Deschmann, Victor U. Weiss

**Affiliations:** 1Institute of Chemical Technologies and Analytics, TU Wien, A-1060 Vienna, Austria; samuele.zoratto@tuwien.ac.at (S.Z.); gernot.friedbacher@tuwien.ac.at (G.F.); martina.marchetti-deschmann@tuwien.ac.at (M.M.-D.); 2Electron Microscopy Facility, Vienna BioCenter Core Facilities GmbH, A-1030 Vienna, Austria; thomas.heuser@vbcf.ac.at; 3Pharmaceutical Sciences, Baxalta Innovations GmbH (Part of Takeda), A-1221 Vienna, Austria; robert.pletzenauer@takeda.com (R.P.); michael.graninger@takeda.com (M.G.)

**Keywords:** AAV8, VLP, nES GEMMA, DMA, cryo-TEM, gene therapy

## Abstract

Gas-phase electrophoresis on a nano-Electrospray Gas-phase Electrophoretic Mobility Molecular Analyzer (nES GEMMA) separates single-charged, native analytes according to the surface-dry particle size. A volatile electrolyte, often ammonium acetate, is a prerequisite for electrospraying. Over the years, nES GEMMA has demonstrated its unique capability to investigate (bio-)nanoparticle containing samples in respect to composition, analyte size, size distribution, and particle numbers. Virus-like particles (VLPs), being non-infectious vectors, are often employed for gene therapy applications. Focusing on adeno-associated virus 8 (AAV8) based VLPs, we investigated the response of these bionanoparticles to pH changes via nES GEMMA as ammonium acetate is known to exhibit these changes upon electrospraying. Indeed, slight yet significant differences in VLP diameters in relation to pH changes are found between empty and DNA-cargo-filled assemblies. Additionally, filled VLPs exhibit aggregation in dependence on the applied electrolyte’s pH, as corroborated by atomic force microscopy. In contrast, cryogenic transmission electron microscopy did not relate to changes in the overall particle size but in the substantial particle’s shape based on cargo conditions. Overall, we conclude that for VLP characterization, the pH of the applied electrolyte solution has to be closely monitored, as variations in pH might account for drastic changes in particles and VLP behavior. Likewise, extrapolation of VLP behavior from empty to filled particles has to be carried out with caution.

## 1. Introduction

Gene therapy alters the genetic information of an organism, replacing, supplementing, or modifying its genomic profile to enable hereditary disease treatment, as lately reviewed for hemophilia [1], for example. To convey new genomic information to a cell, cargo carriers are necessary to shield nucleotides from the environment and achieve their targeted transport. One possible carrier type is virus-like particles (VLPs), macromolecular assemblies resembling their parent virus but no longer being infectious due to a lack of viral genomic information [2]. Hence, VLPs can be engineered to encapsulate other genomic information in question.

AAV are viruses of the genus *Dependoparvovirus* within the family *Parvoviridae*. They are approximately 26 nm in size, non-enveloped, non-pathogenic, and endemic in humans as well as several vertebrate species. To date, 13 serotypes have been identified in nature, each with a different tissue tropism. The capsid comprises 60 copies of three types of subunits: VP1, VP2, and VP3, arranged in a T = 1 icosahedral symmetry with a molar ratio of 1:1:10. Due to its small size, only 4.7 kb of single-strand DNA (ssDNA) can be encapsulated [3]. 

In this work, we took an interest in adeno-associated virus serotype 8 (AAV8) modified to generate VLPs for gene therapy applications. AAV8 VLPs were provided in two different preparations: (i) lacking any genomic cargo (i.e., an ‘empty’ VLP preparation), and (ii) carrying a non-viral engineered genome (i.e., a ‘filled’ VLP preparation).

A possible technique for (bio-)nanoparticle characterization in general and AAV8 in particular is gas-phase electrophoresis on a nano Electrospray Gas-phase Electrophoretic Mobility Molecular Analyzer (nES GEMMA), also known as nES Differential Mobility Analyzer (nES-DMA) [4]. This technique is based on the size separation of single-charged, surface-dry analytes in the gas phase. Transfer of analytes from a liquid sample—a volatile electrolyte solution is a necessary prerequisite for the technique—to the gas-phase is achieved via a nES process, followed by drying of droplets and concomitant charge equilibration of particles in a bipolar atmosphere induced by, for example, a ^210^Po α-particle source, a soft X-ray charger, or an alternating corona discharge process [5,6,7,8]. The majority of particles lose all charges upon passage of the bipolar atmosphere and are not regarded further, whereas a certain percentage of aerosolized analytes remains single-charged and are introduced to the DMA unit of the instrumentation. There, two forces act on the particles during size separation. Firstly, analytes are transported via a high laminar sheath flow of compressed, filtered ambient air through the DMA. At the same time, a tunable electric field exerts a force on charged particles in an orthogonal direction. By variation of the applied field strength and based on electrophoretic principles, only particles of a corresponding surface dry particle diameter (electrophoretic mobility, EM diameter) have the correct trajectory to continue to the detector unit of the instrument. There, particles are counted after a nucleation process in a supersaturated atmosphere as they pass a focused laser beam. This setup guarantees true particle-number-based detection of smaller-sized sample components next to larger ones in accordance with the recommendations of the European Commission (2011/696/EU, 18 October 2011, updated 2022) for nanoparticle characterization. The same instrumentation is also known under several other names, e.g., nES-DMA, macroIMS, SMPS or liquiScan ES (e.g., [9,10]).

Since its first reference in the literature by Kaufman et al. in 1996 [4], gas-phase electrophoresis has been applied for the characterization of a multitude of (bio-)nanoparticle materials, such as viruses [11,12,13], VLPs [14,15,16,17], liposomes and lipoprotein particles [18,19,20,21,22], extracellular vesicles [23,24,25], and organic and inorganic nanoparticles [26,27,28]. Frequently, ammonium acetate is employed as an electrolyte solution for the nES process. However, ammonium acetate is no buffer per definition (i.e., ammonium acetate solution), and especially at neutral pH, it exhibits no buffering capacity. Hence, significant changes in pH occur upon drying of droplets generated in the nES process, as reported by Konermann in 2017 [29]. We now took interest in the question how AAV8-based VLPs would behave to such pH changes, focusing on a well-defined in-solution setup. Analyses were carried out via gas-phase electrophoresis, and we corroborated our findings by atomic force microscopy (AFM) and cryogenic transmission electron microscopy (cryo-TEM) as orthogonal analytical techniques. All applied analytical techniques relate a pH dependency of AAV8 particles in aqueous ammonium acetate. 

## 2. Materials and Methods

### 2.1. Chemicals

Ammonium acetate (NH_4_OAc, ≥99.99%), ammonium hydroxide (ACS reagent), and acetic acid (ACS reagent) were purchased from Sigma-Aldrich (Steinheim, Germany). The electrolyte solution was prepared by dissolving 40 mM of ammonium acetate with water of ultrahigh quality (UHQ) delivered by a Simplicity UV apparatus (18.2 MΩ × cm at 25 °C, Millipore, Billerica, MA, USA). The solution was adjusted to different pH levels ranging from 4.0 to 9.0 (with 1.0 step increase) with ammonium hydroxide or acetic acid. Lastly, the solution was filtered through a surfactant-free cellulose acetate membrane with 0.20 µm pore size syringe filters (Sartorius, Göttingen, Germany).

### 2.2. Sample Description

Purified AAV8 VLP samples were provided by Baxalta Innovations GmbH (Orth/Donau, Austria, part of Takeda). Two different batches were provided: (i) so-called empty AAV8 VLPs (3776 µg/mL, i.e., 7.3 × 10^14^ capsids/mL) with 97% of capsids not carrying any genomic information, and (ii) so-called filled AAV8 VLPs (85 µg/mL, i.e., 1.6 × 10^13^ capsids/mL), where 66% of all the capsids were carrying a genomic load. The percentage of capsid filling was assessed via cryogenic transmission electron microscopy (cryo-TEM) by Baxalta Innovations GmbH.

### 2.3. Sample Preparation

Buffer exchange against 40 mM NH_4_OAc at pH 7.0 was carried out by means of 10 kDa MWCO centrifugal filters (polyethersulfone membrane from VWR, Vienna, Austria). After three repetitions of spin filtration at 9000 g each, the retentate (approx. 10 µL in volume) was reconstituted in a pH-adjusted electrolyte solution (i.e., ranging from 4.0 to 9.0) to yield 22 µg/mL and 8.5 µg/mL final concentration for empty and filled AAV8 VLPs, respectively. Before analysis, the samples were further diluted to yield 250–300 particle counts for the signal at 25 nm at pH 7.0 with nES GEMMA. The respective dilution ratio was then employed for AFM and cryo-TEM analysis.

### 2.4. Instrumentation

nES GEMMA analyses were carried out on a TSI Inc instrument (Shoreview, MN, USA), which consisted of a nanoelectrospray charge reduction source unit (model 3480) equipped with a ^210^Po charge equilibration device, an electrostatic classifier control unit employing a nano differential mass analyzer (nano DMA; model 3080) and an n-butanol driven ultrafine condensation particle counter (CPC; model 3025A) for AAV8 VLP detection. The nES unit is equipped with a 24 cm long polyimide-coated fused-silica capillary with an inner diameter of 25 µm (Molex, Lincolnshire, IL, USA). The capillary is manually cut and tapered with a home-built grinding machine based on the work of Tycova et al. [30].

Nanoparticle separation and detection were achieved with the following settings: the filtered airflow on the nES generator was set to 1.6 × 10^−5^ m^3^/s (1 L per minute, Lpm), the CO_2_ gas flow to 1.6 × 10^−6^ m^3^/s (0.1 Lpm, 99.5% from Messer, Gumpoldskirchen, Austria) and the differential capillary pressure at 27.58 kPa (4 pounds per square inch differential, PSID). Capillary conditioning was performed by pre-spraying each sample for at least 3 min before starting the measurement. Capillary rinsing was performed by infusing the electrolyte solution until the signal from the previous sample was no longer detectable. The sample was infused at a flow rate of 70 nL/min. The capillary tip voltage was set to have a stable Taylor cone (approx. 2 kV voltage and −380 nA current). The electrostatic classifier was set in automatic scanning mode (up scan time 120 s, retrace time 30 s) with a sheath gas flow rate of 2.5 × 10^−4^ m^3^/s (15 Lpm), which yielded a range of measurable EM diameters between 1.95 nm and 64.9 nm. A total of 10 scans for each sample were used to generate a median spectrum. Mathematical and statistical calculations on the nES GEMMA spectra were made with the software OriginPro ver. 2021 (OriginLab Corporation, Northampton, MA, USA).

AFM experiments were carried out on a NanoScope VIII Multimode SPM instrument (Bruker, Bremen, Germany) using silicon cantilevers with integrated silicon tips (model NCHV from Bruker, resonance frequency: 320 kHz, L = 125 µm, k = 42 N/m). The images were acquired in tapping, constant amplitude mode at a scanning rate of 1.99 Hz over a scan area of 5 and 1 µm^2^. 

The homogeneity of the mica platelet was tested prior to sample analysis. Subsequently, 10–20 µL of sample (5–20 µg/mL solutions) were spotted on the platelet’s surface at room temperature. The sample was allowed to adsorb for 5 min undisturbed before being gently rinsed with UHQ water and dried with nitrogen gas (outlet pressure 350 kPa).

Lastly, the AFM images have been analyzed by NanoScope Analysis 1.5 software (Bruker, Santa Barbara, CA, USA). Longitudinal and medial plane sections were used to generate profiles of the imaged particles.

Cryo-TEM samples were prepared using Quantifoil (Großlöbichau, Germany) Cu 200 mesh R2/2 holy carbon grids which were glow discharged for 60 s at −25 mA with a Bal-Tec (Balzers, Liechtenstein) SCD005 glow discharger. Grids were mounted on forceps and loaded into a Leica GP (Leica Microsystems, Vienna, Austria) grid plunger with the climate chamber set to 4 °C and 75% relative humidity. Sample aliquots of 4 µL were applied to the carbon side of the grid and front-side blotted for 2 s with Whatman filter paper #1 (Little Chalfont, Great Britain) and plunge frozen into liquid ethane at approximately −180 °C for instant vitrification. Cryo-samples were transferred to a Glacios cryo-transmission microscope (Thermo Scientific, Waltham, MA, USA) equipped with a X-FEG and a Falcon3 direct electron detector. The microscope was operated in low-dose mode and digital images were recorded using the SerialEM software [31] and the Falcon3 camera in linear mode at a defocus of −6 µm and 36,000 fold magnification corresponding to a pixel size of 4.124 Å.

### 2.5. Cryo-TEM Particles Diameter Determination 

Individual VLP particles were picked manually from suitable cryo-TEM micrographs and their coordinates were saved as model points using the open-source IMOD processing tool [32]. The particle models were used for alignment and 2D averaging with the Particle Estimation for Electron Tomography (PEET) software, an open-source tool typically used for 3D subtomogram averaging [33,34]. Unbiased diameter determination was achieved by an in-house ad hoc developed Python script (see Appendix A), able to generate 3D graphs from the 2D images; by plotting correlating grey values of each (*x*,*y*) image’s pixel coordinates. The 3D graphs can be sectioned and exported at different height levels. For each sample, three sections at different levels were extracted. At last, the data points contained in the sections were imported to OriginPro software (ver. 2021), where a circle fitting function was applied to determine the diameter of the particle.

## 3. Results

Our article aims to expand the knowledge concerning the characteristics and behavior of AAV8-based VLPs suspended in electrolyte solutions at different pH levels. nES GEMMA technology is used as main analytical technique. As previously introduced, nES GEMMA can be easily employed to analyze viruses and deliver robust, high-throughput, and cost-effective analyses. Moreover, orthogonal analytical techniques such as AFM in tapping mode and cryo-TEM have been applied to corroborate our results obtained from gas-phase electrophoresis.

As already demonstrated in our previous work [35], nES GEMMA can discriminate between VLPs either carrying or lacking genomic information for the two individual sample types (Figure 1A). The small, albeit critical, difference in EM diameter between the two VLP preparations originates from the presence, or absence, of an encapsulated cargo (i.e., the genomic material). The discrimination is possible due to the operating conditions of nES GEMMA analysis, which induce evaporation of the hydrodynamic layer surrounding the particles, thus yielding surface-dry analytes. However, the surface drying effect seemingly propagates across the proteinaceous layer of the capsid to its core, as the VLP EM diameter is affected by the content of its cargo. Two possible outcomes can be surmised: in the presence of a payload, the particles mostly retain their original shape thanks to stabilizing forces emerging from protein-genome interactions. In the absence of a genomic load instead, a higher degree of deformation is allowed, thus making the particles more flexible and detectable at smaller EM diameters.

A mandatory step for nES GEMMA includes the exchange of the original sample buffer to a volatile electrolyte solution. This drastically reduces non-volatile components, which hinder correct and accurate particle EM diameter determination. More details can be found in previous works [36,37]. To some extent, either for nES GEMMA or for ESI MS analysis, aqueous ammonium acetate is applied in that context. However, it has to be considered that ammonium acetate droplets might undergo drastic pH changes upon evaporation, as discussed by Konermann in detail [29]. Thus, we took an interest in investigating the effects of different pH levels on VLP preparations, either carrying a genomic cargo or not, and both suspended in aqueous ammonium acetate. Results are displayed in Figure 1B and Figure 2A, respectively. 

### 3.1. AAV8 VLPs in Aqueous Ammonium Acetate at Different pH Targeted by Gas-Phase Electrophoresis

Thanks to their distinct size differences, VLP preparations, either carrying or lacking cargo, can be easily discriminated (*p* < 0.0001, Figure 1B). In detail, filled VLPs have a constant higher EM diameter, and show a minimal downward trend towards a smaller EM diameter for alkaline pH. For empty VLPs instead, a sharper pH-dependent trend towards a smaller EM diameter is observed. Reasonably, pH-driven protein modification alters capsid’s structural stiffness in both VLP preparations, but empty ones are again more affected. 

In addition, below a physiological pH level, a significant contrast in the number of detected particles between filled and empty VLP preparation is noticed (*p* < 0.0001, Figure 2A). A high detection yield for empty VLPs confirms that nES GEMMA capabilities are not hindered in an acidic working environment; hence, the observed difference for VLP preparations is sample-dependent. The unique behavior demonstrated by filled VLPs is of great importance, both for VLP analysis and in the biopharmaceutical field during manufacturing steps. In vivo, viruses commonly hijack the cellular uptake mechanism to infect the cells. A number of them take advantage of the acidic environment of late endosomes and endolysosomes to shed their proteinaceous layer to deliver their naked viral load directly into the cytoplasm. Other viruses instead—AAV8 included—need to survive and accomplish endosomal escape intact in order to complete the infection [38,39]. Therefore, it is unlikely that bionanoparticles depletion is caused by viral uncoating. Moreover, capsid fragments would be detected at EM diameters below 25 nm; still, the nES GEMMA signals in this region are negligible and comparable to empty VLP preparations at comparable pH levels. Another viable option is aggregation, where filled VLPs at acidic pH generate clusters outside the operating range of nES GEMMA. To investigate this claim, filled VLPs in acidic conditions have been analyzed via AFM. In fact, VLP aggregates sized >200 nm are clearly visible in the AFM images (Figure 2B(i)). AFM images at higher magnification (Figure 2B(ii)) clearly show aggregate species stemming from monomeric particles. Longitudinal and medial sections of the imaged species match profiles fitting 25 nm monomeric particles (Appendix A). It is of note that particle aggregation yielding such large assemblies cannot be followed by the applied nES GEMMA setup. Furthermore, it has to be stressed that the resulting pH-dependant aggregation is partially reversible as adjusting the pH back to a neutral value restores the corresponding monomer peak in nES GEMMA measurements (Appendix A). 

As demonstrated, the presence or absence of an encapsulated cargo significantly alters the capsid’s structural characteristics. The novel behavior observed in filled VLPs additionally shows aggregation once exposed to acidic conditions. This pH-driven phenomenon is likely caused by a newly acquired feature linked to an increase in the capsid-to-capsid adherence with or without DNA mediation, and might be caused by fine capsid’s structural changes. Although nES GEMMA can determine particles’ diameter with both high accuracy and precision with high sample throughput times and good statistics, it can only partially infer the shape details of the sample. For this purpose, analytical techniques capable of imaging at near-atomic levels, such as cryo-TEM, are more fitted, although very time-consuming. Therefore, VLP preparations at both pH extremes (i.e., pH 4 and 9) have been analyzed with cryo-TEM technology to investigate if any distinct features would arise between samples (Figure 3).

### 3.2. Cryogenic Transmission Electron Microscopy (cryo-TEM) of AAV8 VLPs

Cryo-TEM RAW images have low signal-to-noise contrast between the investigated material and the background. Hence, averaging with software is necessary to increase contrast and highlight the sample’s characteristics. In our work, we employed the Particle Estimation for Electron Tomography (PEET) open-source software to catalog, align, and average hundreds of particles per sample (Figure 3A). Originally designed as an averaging program for 3D tomograms, we used it for 2D averaging. To ensure a reliable, biased-free diameter determination, the averages generated by PEET were processed via a Python script written in-house (included in the Appendix A), where sections at different height were extracted. The data points included in these sections were later fitted with a circle function via the OriginPro (ver. 2021) software. This workflow, providing unbiased, reliable data for PEET-generated cryo-TEM images of VLPs, is summarized in Figure 4 and could be applied to images generated from different analytical techniques.

Visual comparison of the PEET-generated Cryo-TEM images show remarkable structural differences (Figure 3A). Within the filled VLP preparation, the sample at pH 4 (Figure 3A(i)) shows a well-defined icosahedral capsid’s structure with regular, straight edges, and clear vertices; at pH 9 (Figure 3A(ii)) the same characteristics are instead less pronounced. Whereas in the empty VLP preparation, regardless of the pH value, the capsid’s structure appears smoother, rounder, and slightly smaller (Figure 3A(iii,iv)). Anticipated by nES GEMMA results, and now corroborated by cryo-TEM images, the particle’s shape is linked with the presence or absence of an encapsulated cargo, and this could affect the capsid’s interactions at the macromolecular scale. 

Diameter determination results in filled VLPs exhibiting larger diameters than empty ones, in agreement with nES GEMMA observations (*p* < 0.001, Figure 3B). Moreover, new information about the sample’s characteristics can be inferred. In detail, empty VLPs report identical diameter values regardless of the solvent’s pH, which is contrary to the downward trend observed with nES GEMMA. In hydrated conditions, regardless of the pH, empty VLPs maintain a constant size and shape. Filled VLPs instead show a small diameter variability between pH conditions, probably due to the sharper contours as reflected in the imaged particles (Figure 3A(i,ii)). Nonetheless, this slight visual and diameter difference does not provide any functional proof to explain the observed aggregation behavior.

## 4. Discussion

In this study, we demonstrated the feasibility of nES GEMMA to discriminate, detect, and characterize novel behaviors of AAV8-based VLP preparations analyzed with nES GEMMA-compatible electrolyte solution adjusted to different pHs. Moreover, to observe the VLP preparations in their hydrated native state, cryo-TEM analysis was performed.

Results from nES GEMMA of the two VLP preparations show peculiar, unique characteristics. Filled VLPs report limited EM diameter deviation across the different pHs, but detection efficiency is hindered in acidic conditions. Instead, the particle detection efficiency of empty VLPs is unperturbed by the electrolyte’s pH, but pH-depended EM diameter variability is observed. 

Results from PEET-based cryo-TEM images highlight the structural differences between the two VLP preparations. Moreover, the developed method for biased-free diameter determination via cryo-TEM confirms size-specific differences between empty and filled VLP preparations as observed with nES GEMMA. However, within VLP preparations, pH-dependent variation is observed, contrasting nES GEMMA results—specifically, a constant diameter for empty VLPs and a variable one for filled VLPs. One hypothesis to explain the discrepancy observed between nES GEMMA and cryo-TEM resides in the different working environments of these techniques. In nES GEMMA, the particles are sorted and classified in a solvent-free environment and are in a surface-dry state. Consequently, empty VLPs, lacking an encapsulated cargo, collapse due to structural stress caused by solvent evaporation. Likely, the solvent’s pH alters protein conformation, which translates to different degrees of structural stability or stiffness. Therefore, labile structures are detected at smaller EM diameters. Instead, filled VLPs remain unaffected, thanks to the incompressible characteristics of their payload. On the contrary, in cryo-TEM, the VLPs are analyzed in a solubilized native state. Indeed, empty VLPs are not subjected to structural stress from solvent evaporation, and maintain their original form. Filled VLPs instead show a slight difference between the two pH extremes. The origin of this difference is as yet unclear. Putatively, and in addition to pH-induced changes at the protein level in acidic conditions, an osmotic pressure difference is generated at the semipermeable membrane level (i.e., proteinaceous capsid layer) caused by the negative charges of the genomic cargo. Hence, solvent intake is favored to equilibrate the osmotic pressure, causing the particle to swell. 

In acidic conditions, nES GEMMA reveals a steep decrease in particle counts for filled VLPs, while empty ones are unaffected. Indeed, AFM analysis of filled VLPs in acidic conditions confirmed our hypothesis: the lower particle count was due to particle aggregation. Although nES GEMMA’s working range of detection could be expanded, the heterogeneity of these clusters impedes their detection without an upstream enrichment technique, as observed in previous work [36]. Aggregation could be explained by the shape of the particles, highlighted by cryo-TEM-based PEET averaged images. The well-defined capsid faces of filled VLPs might provide the infrastructure for non-covalent capsid-to-capsid interaction, thus supporting aggregation. However, since no aggregate formation was observed in cryo-TEM averages, it suggests that the onset of aggregation requires further contributing factors. Once again, the critical factor might lie in the different working environments, dry in both nES GEMMA and AFM, and wet (i.e., plunge frozen in a solvent) for cryo-TEM. Putatively, if aggregation is promoted in a dry environment, it might be an evolutionary adaptation of AAV to increase viral infection transmission efficiency through airborne routes. 

The results presented in this study demonstrate the feasibility and resilience of nES GEMMA as a method of analysis for AAV Serotype-8 VLP preparations across different pH levels. This is of great importance, especially in overseeing the different phases of a drug production process, where simple, reliable, and inexpensive control is often sought. 

The results observed via AFM and cryo-TEM analysis provided complementary information to investigate the behavior observed with nES GEMMA when filled VLPs are exposed to acidic pH. AFM reveals the formation of heterogeneous aggregate products that could not be directly targeted with nES GEMMA but whose presence could be inferred by the significant drop in particle counts in the EM diameter range characteristic of VLPs in a monomeric state. Initially employed to investigate particle aggregation, cryo-TEM analysis did not confirm cluster formation observation. However, its extreme magnification power provided valuable information, and its application will propel future development toward the characterization of VLPs.

To conclude, the developed method and analytical approaches presented in this study provided new insight and possible applications for both the scientific and the biopharmaceutical field. The observations herein open several discussion points, encouraging further studies to investigate such valuable drug-delivery vectors.

## Figures and Tables

**Figure 1 viruses-15-01361-f001:**
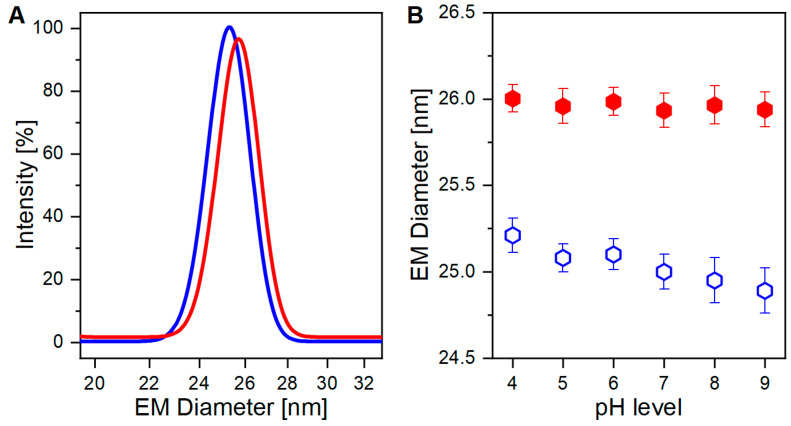
**nES GEMMA analysis of empty and filled VLP preparations.** (**A**) Gaussian fitting overlay of empty (blue trace) and filled (red trace) VLPs at pH 7. (Adapted from Zoratto et al., 2021 [35]) (**B**) EM diameter changes of empty (empty blue hexagons) and filled (filled red hexagons) VLPs according to pH levels (*n* = 6 analyses per pH level, average EM diameters and standard deviations are depicted, *p* < 0.0001).

**Figure 2 viruses-15-01361-f002:**
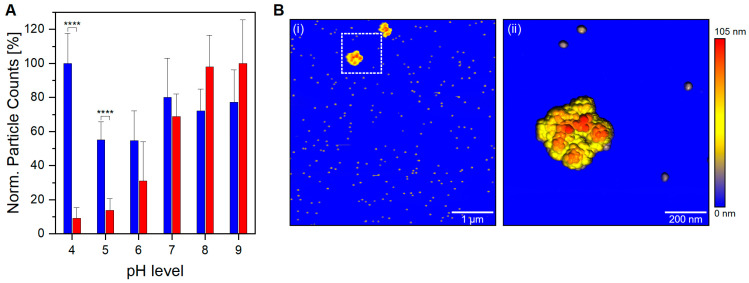
**Behavior of VLP preparations at different pH levels.** (**A**) Normalized particle counts detected via nES GEMMA for empty (blue columns) and filled (red columns) preparations at different pH levels (*n* = 6 measurements; average values and standard deviations are depicted; values at a corresponding pH are related to particle counts at pH 4 for empty VLPs and pH 9 for filled VLPs). For filled VLPs a decline in particle numbers with decreasing pH is observed in contrast to empty particles. (**B**) AFM analysis of filled VLPs at pH 5; (**i**) the analyzed area shows monomers and two aggregate clusters; (**ii**) magnification of the white dotted area in (**i**). **** *p*  <  0.0001.

**Figure 3 viruses-15-01361-f003:**
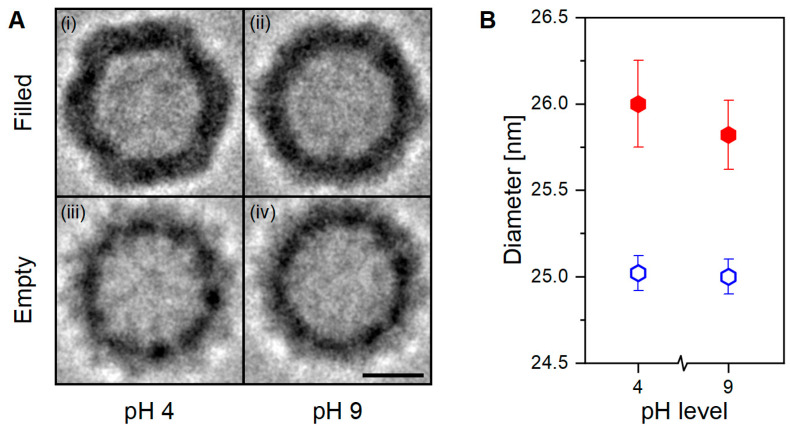
**Cryo-TEM analysis of empty and filled AAV8 VLPs.** (**A**) AAV8 VLPs filled at pH 4 (**i**) and pH 9 (**ii**), and empty at pH 4 (**iii**) and pH 9 (**iv**) after 2D averaging. (**B**) Particles’ diameter determined via Python script from the aligned particles showed in (**A**). Black bar equals 10 nm.

**Figure 4 viruses-15-01361-f004:**
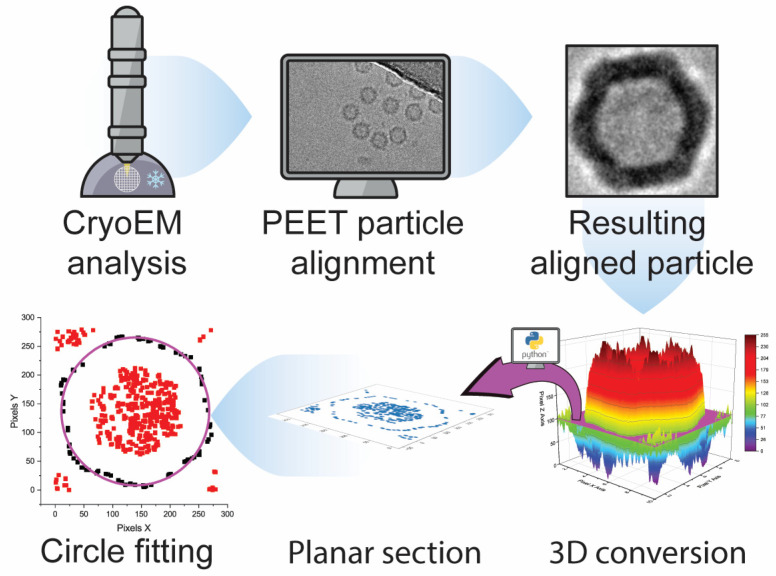
**Workflow scheme from cryo-TEM analysis to biased-free diameter determination.** In the last tile, the magenta circle represents the fitted function applied to the black data points. Red data points are instead excluded from the fitting process.

## Data Availability

Data is available upon request.

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
