# Peer review of "Adeno-Associated Virus-like Particles’ Response to pH Changes as Revealed by nES-DMA"

_viruses, 2023, doi:10.3390/v15061361_

Round 1

Reviewer 1 Report

Summary:

Zoratto et al. examine how gas-phase electrophoresis on nES GEMMA can separate native analytes based on particle size. The study focuses on adeno-associated virus 8 (AAV8) based virus-like particles (VLPs) used in gene therapy and shows that the pH of the electrolyte solution affects the size and behavior of the VLPs. Variations in pH of the electrolyte solution should be closely monitored when characterizing VLPs.

General comments:

In general, the authors introduced a novel method to characterize VLPs and showed the characteristics of the VLPs change upon the PH change. However, more evidence as well as the rationale behind the experimental design would need to be provided to back the authors’ claim and to be considered for publication in Viruses.

Specific Comments:

The claim about the ‘behavior’ of the VLP is misleading given no functional feature has been described across the paper.

The Cryo-TEM analysis needs to be done with different PH condition.

For all data presented, given the low N number and the small effect, individual points should be plotted.

Author Response

Please refer to pdf file for reply.

Reviewer 2 Report

  Review of MS viruses-2405767:

Adeno-associated virus-like particles’ response to pH changes as revealed by nES-DMA, by  Samuele Zoratto, Thomas Heuser, Gernot Friedbacher, Robert Pletzenauer, Michael Graninger, Martina Marchetti-Deschmann and Victor U. Weiss

This study extends a prior investigation of AAV particles by the same research group, where mobility analysis in the gas phase had shown slight, though nevertheless observable, mobility differences between purified empty and full particles of AAV8 (their reference 30).[1] The new contribution under review explores the role of solution pH. The full particle is seen to have a mobility diameter unaffected by pH, while its abundance decays at low pH. The abundance of the empty particle is independent of pH, while its size decreases slightly at increasing pH. The results from the mobility study is complemented with cryo-electron microscopy imaging, which gives dimensions comparable to measured mobility diameters for both empty and full capsids.

These findings are of substantial scientific interest, and should be published after minor improvements are implemented.

1) The following statement is made: “As already demonstrated in our previous work [30], nES GEMMA can discriminate between VLPs either carrying or lacking genomic information (Figure 1A)”. However, the ability to distinguish between empty and full capsids by mobility alone depends not only on the difference in mobility (or diameter), but also on the widths of the two mobility peaks. Please, discuss the confidence with which the concentrations of the two contributions could be inferred in a real situation where both particles would be present under comparable abundances. Since the difference in mean diameter increases with increasing pH, it would seem that the discrimination would be easier at larger pH. But this may not be the case if the peak width of either species also increases with pH. This discussion should accordingly be focused on all the range of pH values investigated.   

2) Are the width of the peaks for the empty and filled capsids different? Or is this parameter not determined by the particles, but rather by the finite DMA resolution?

3) In prior studies from this group, 1,[2] the issue of distinguishing empty and filled capsids was considered more generally for a number of different species, and correlations between mass and mobility diameter were provided for both empty and filled particles. However, points (1) and (2) above were not entirely clear relating to these other viruses. It would be most helpful to other researchers in this field if the new article would address these issues more generally, extending the discussion to as many viruses for which empty capsids have been characterized by gas phase mobility.

[1] Zoratto, S.; Weiss, V. U.; van der Horst, J.; Commandeur, J.; Buengener, C.; Foettinger-Vacha, A.; Pletzenauer, R.; 457 Graninger, M.; Allmaier, G., Molecular weight determination of adeno-associate virus serotype 8 virus-like particle either 458 carrying or lacking genome via native nES gas-phase electrophoretic molecular mobility analysis and nESI QRTOF mass 459 spectrometry. J Mass Spectrom 2021, 56 (11). 460

[2] Victor U. Weiss, Ronja Pogan, Samuele Zoratto, Kevin M. Bond, Pascale Boulanger, Martin F. Jarrold, Nicholas Lyktey, Dominik Pahl, Nicole Puffler, Mario Schelhaas, Ekaterina Selivanovitch, Charlotte Uetrecht, Günter Allmaier, Virus-like particle size and molecular weight/mass determination applying gas-phase electrophoresis (native nES GEMMA), Analytical and Bioanalytical Chemistry (2019) 411:5951–5962

.

adequate

Author Response

Please refer to pdf file for reply.

Reviewer 3 Report

In this study, the authors employed the nES GEMMA approach to explore the behaviors of AAV8 vectors under various pH conditions, supplemented by AFM and cryo-TEM. I have a few suggestions for improvement:

  1. The discussion could benefit from more comprehensive details about how alterations in pH affect the conformation of the AAV capsid protein. Are changes in shape or diameter observed? Previous literature indicates that shifts in pH facilitate AAV's escape from the endosome during infection.
  2. The removal of empty AAV is a critical step in AAV purification for gene therapy. It would be beneficial to know the most commonly used method in the industry to distinguish between full and empty AAV particles. Are the methods used in this study (nES GEMMA, AFM, and cryo-TEM) capable of identifying full capsids for gene therapy? What are the pros and cons of these methods?
  3. The authors state, "Indeed, slight, yet significant differences in VLP diameters in relation to pH changes are found between empty and DNA-cargo-filled assemblies." It would be useful to see the statistical significance analysis in Figures 1B, 2A, and 3B to support this claim.
  4. Given that the AAV capsid protein bears more negative charges than positive ones, the overall viral particle typically exhibits net negative charges. According to Figure 2A, full AAVs seem to aggregate at low pH, possibly because the acidic environmental pH is close to the isoelectric point (pI). How do the authors explain the different patterns shown by empty particles compared to full particles in Figure 2A?
  5. The significance of this study is not entirely clear. Is the focus on the changes in AAV conformation at different pH levels? Or on the advantages of detection methods? What potential applications can this study inform?

Author Response

Please refer to pdf file for reply.

Reviewer 4 Report

This manuscript presents a careful analysis of physical properties of empty and filled AAV particles, using unusual techniques including gas-phase electrophoresis. The data show that the empty particles are somewhat smaller than filled particles and that the filled particles tend to “disappear” at low pH. In general the findings are clear and of interest. One point I found difficult to understand was the data presented in Fig. 2A; I would suggest an additional explanatory sentence explicitly describing what is presented there and how these numbers were obtained.

Author Response

Please refer to pdf file for reply.

Round 2

Reviewer 1 Report

I appreciate the authors' effort in responding to my comment.

Reviewer 3 Report

Thanks for addressing my comments. However, the authors still have not address the statistic analysis issue in figures. These must be done before publication because the authors claim "significant" in the manuscript. The big number of samples should not be the obstacle of statistic analysis. 

Author Response

Dear Ms. Nova Peng, dear reviewer,

we thank the referee for his/her comment. The calculated p-value scores have now been included in the manuscript in order to underscore our claim of 'significant differences' between empty and filled VLP particles. We think that our manuscript is now fit for publication and thank all the reviewers for their valuable input.

Kind regards,

Victor Weiss and Samuele Zoratto